# Comparison of Various Reducing Agents for Methane Production by *Methanothermobacter marburgensis*

**DOI:** 10.3390/microorganisms11102533

**Published:** 2023-10-10

**Authors:** Maximilian Peter Mock, Rayen Ochi, Maria Bieringer, Tim Bieringer, Raimund Brotsack, Stephan Leyer

**Affiliations:** 1Technology Centre Energy, University of Applied Sciences Landshut, Wiesenweg 1, 94099 Ruhstorf an der Rott, Germany; maximilian.mock@haw-landshut.de (M.P.M.);; 2Department of Engineering, Faculty of Science, Technology and Medicine, University of Luxembourg, 1359 Luxembourg, Luxembourg; stephan.leyer@uni.lu; 3European Campus Rottal-Inn, Deggendorf Institut of Technology, Max-Breiherr-Straße 32, 84347 Pfarrkirchen, Germany

**Keywords:** biological methanation, anaerobic media, reducing agent, sodium dithionite, sodium sulfide, L-cysteine-HCl, *Methanothermobacter marburgensis*

## Abstract

Biological methanation is driven by anaerobic methanogenic archaea, cultivated in different media, which consist of multiple macro and micro nutrients. In addition, a reducing agent is needed to lower the oxidation–reduction potential (ORP) and enable the growth of oxygen-sensitive organisms. Until now, sodium sulfide (Na_2_S) has been used mainly for this purpose based on earlier published articles at the beginning of anaerobic microbiology research. In a continuation of earlier investigations, in this study, the usage of alternative reducing agents like sodium dithionite (Na_2_S_2_O_4_) and L-Cysteine-HCl shows that similar results can be obtained with fewer environmental and hazardous impacts. Therefore, a newly developed comparison method was used for the cultivation of *Methanothermobacter marburgensis*. The median methane evolution rate (MER) for the alternatives was similar compared to Na_2_S at different concentrations (0.5, 0.25 and 0.1 g/L). However, the use of 0.25 g/L Na_2_S_2_O_4_ or 0.1 g/L L-Cys-HCl led to stable MER values over consecutive batches compared to Na_2_S. It was also shown that a lower concentration of reducing agent leads to a higher MER. In conclusion, Na_2_S_2_O_4_ or L-Cys-HCl can be used as a non-corrosive and non-toxic reducing agent for ex situ biological methanation. Economically, Na_2_S_2_O_4_ is cheaper, which is particularly interesting for scale-up purposes.

## 1. Introduction

Biological methanation (BM) plays an important role in improving the energy independence of Europe by providing sustainable and decarbonized biomethane [1]. For example, the IRENA report on “Bioenergy for energy transition” states that the enhanced use of biomass materials and energy is an important factor in meeting the 1.5 °C scenario. In addition, the combination of bioenergy with carbon capture and storage (BECCS) may further reduce global greenhouse gas emissions. It may even provide negative emissions, as previously biologically bound carbon dioxide is captured and stored [2]. The most known example of bioenergy is biogas from anaerobic fermentation, containing around 45–65% methane and 35–55% carbon dioxide, with traces of ammonia and hydrogen sulfide. However, the injection of biogas into the existing gas grid is restricted in terms of different contaminants, which need to be separated [3]. BM is capable of converting the remaining CO_2_ and upgrading biogas to biomethane, which can be injected and distributed throughout the European gas grid. For this purpose, different methanogenic archaea are used, which were found and investigated in the last century.

*Methanothermobacter marburgensis* is the most widely used methanogenic archaeon for BM, which was first isolated from sewage sludge in Marburg (strain Marburg$H^{T}$ [4]), as well as various other locations [5,6]. The organism was identified as a variant of the previously found *Methanobacterium thermoautotrophicum $\Delta H^{T}$*, isolated from sewage sludge in Urbana, Illinois [7]. Both organisms were investigated using DNA–DNA hybridization by Brandis et al. They concluded that these strains are not closely related but the differences are not evident enough to generate a new species [8]. However, Wasserfallen et al. showed with three independent datasets (16S rRNA sequences, antigenic fingerprinting, and plasmid and phage typing) that the strains are different from each other and described the new genus *Methanothermobacter* with the respective species, which was also proposed by Boone et al. [9,10].

The isolation and cultivation of anaerobic methanogenic archaea like *M. marburgensis* is difficult. Special care should be taken to ensure that the medium used meets the nutrient requirements of the organism, which correspond to the environment prior to isolation and are based on the elemental composition of the cell. Furthermore, oxygen can react with substances in cells and inhibit growth or increase toxicity to organisms. Mylroie and Hungate verified in different experiments that a low ORP is needed for the growth of *M. formicicum* [11]. To simplify cultivation under oxygen-free conditions, Hungate developed the so-called roll tube technique in 1950 [12], which was further improved [13] but also adapted for different applications by various researchers. One example is the commonly used methanogenic bacteria cultivation technique by Balch and Wolfe [14]. In order to bind remaining oxygen, the medium is then reduced by the addition of different suitable chemicals (see Table 1).

## 2. State of the Art

In the past, different reducing agents were determined and compared. In 1954, Mylroie and Hungate conducted experiments with *M. formicicum* using sodium sulfide as a reducing agent for the medium. They found that substitution with cysteine did not improve the results [11]. In 1961, Bryant et al. reported that the usage of sodium sulfide instead of cysteine, which was commonly used at this time, led to a greater growth of ruminal bacteria. They also stated that the use of other reducing agents, including dithionite, was not satisfactory for the cultivation of such organisms [25]. Later on, Hungate published an article about his roll-tube technique in 1969, for which he used several reducing agents to adjust the ORP of the media. He found that hydrogen sulfide, which is the product of sodium sulfide at pHs of 6–7, may be the best option as a reducing agent. Furthermore, he stated that the reducing agents had an inhibiting or even toxic effect at higher concentrations. This applies in particular to sodium dithionite [12,26]. As *M. marburgensis* was further described and investigated with respect to the growth conditions and trace elements needed for metabolism by Schönheit et al. in 1979 and 1980 [27,28], they also used sodium sulfide as a reducing agent based on the findings mentioned above. Overall, it can be said that based on this research, sodium sulfide was established as reducing agent, whereas other reducing agents like sodium dithionite were not applied.

In contrast, Rothe et al. developed a simpler method in 2000 to cultivate methanogenic and hyperthermophilic anaerobic archaea without the use of sodium sulfide but with sodium sulfite (Na_2_SO_3_). They stated that sodium sulfide causes a series of problems because of its reaction with water and weak acids to hydrogen sulfide. First, in continuous reactors, H_2_S is flushed out of the reactor due to the low solubility in water. Second, the addition of sulfide anions leads to the precipitation of cationic trace elements. Third, precipitated cations hinder cell growth measurement and the distinction between cells and precipitates [29]. At laboratory scale, the amount of hydrogen sulfide produced may be small, and when glass is used, the corrosive properties are neglected. When considering the scale-up and application of high pressures to biological methanation, it is completely different since stainless steel is used for construction. Sulfur compounds such as H_2_S, S_2_O_3_${}^{2-}$, SO_3_${}^{2-}$, HS^−^ and even S_2_O_4_${}^{2-}$ are known to corrode stainless steel. Therefore, specific manufactured stainless steels, e.g., carbon or austenitic stainless steels, are needed for the whole system. However, H_2_S is the most hazardous and toxic chemical compared to the other sulfur compounds [30]. Even small amounts of this chemical can cause several challenges due to technical, economic and obvious safety aspects.

In terms of the state of the art, it is quite common to use sodium sulfide as a reducing agent despite the reported hazardous and environmental disadvantages. Therefore, this study showed that sodium dithionite and L-cysteine-HCl can be used as a substitute reducing agent for *M. marburgensis* with equal or even better methane evolution rates (MER) compared to sodium sulfide. For this purpose, a method was developed to compare and determine the optimal cultivation conditions with respect to MER. The method was established for commonly used 120 mL serum bottles in anaerobic cultivation. The optimal cultivation conditions were then used to compare different reducing agents at different concentrations.

## 3. Materials and Methods

In this chapter, the chemicals, gases, medium and laboratory devices used are described, as well as the setup of the experiments and the applied measurement methods.

### 3.1. Chemicals and Gases

The chemicals used were of analytical grade and purchased from Carl Roth GmbH+ Co. KG Karlsruhe, Germany.For cultivation, a mixture (4:1) of hydrogen (99.999 vol%) and carbon dioxide (99.9 vol%) resulting in 80 vol% hydrogen and 20 vol% carbon dioxide was used. For media preparation and gas chromatography, argon was used as a carrier gas with a purity of 99.999 vol%. All gases were purchased from Westfalen AG, Muenster, Germany.

### 3.2. Growth Medium

For all experiments, *M. marburgensis* was used (DSM 2133), bought from German Collection of Microorganisms and Cell Cultures GmbH, Braunschweig, Germany. The medium was based on Schönheit et al. [27] and was adapted according to Table 2. The trace element solution (TES) was prepared according to recipe 141 (Modified Wolin’s mineral solution) of DSMZ at a ten times higher concentration.

As an indicator for the redox potential in biological systems, sodium resazurin was used. In its inactive form, it has a blue color. When first introduced to a medium near neutral pH, the blue color changes to pink (resorufin) in an irreversible step, usually initiated with heat. However, if resazurin is added to an alkaline solution, such as the hereby used medium with reducing agent, the color may stay blue. In the second, reversible step, the hydroresorufin/resorufin redox couple formed will be pink above −51 mV and colorless below −110 mV [31,32].

### 3.3. Setup Preparation

The cultivation was carried out in serum bottles (120 mL, bought from FloraCura, Garmisch-Partenkirchen, Germany), sealed with butyl rubber stoppers and aluminum crimp caps (both purchased from Ochs Laborfachhandel, Bovenden, Germany). The caps have an opening for injection needles, allowing the extraction of gas or suspension samples. After sealing, the reactor volume V_r_ was 118 mL. For anaerobic handling, an anaerobic chamber (AC, purchased from Toepffer Lab, Adelberg, Germany) was used with an atmosphere consisting of 95 vol% nitrogen and 5 vol.-% hydrogen and a palladium catalyst to regulate the oxygen concentration. For cultivation, a multiposition magnetic stirrer (MIXdrive 6 HT with MIXcontrol eco from 2mag AG, Munich, Germany) was used in an incubator (INCU-Line Prime IL 112, VWR International GmbH, Darmstadt, Germany) at 63 °C, and for each bottle a magnetic stirrer (rod-shaped, 25 × 6 mm) was used.

#### 3.3.1. Parameter Experiment (PE)

The main parameters influencing the MER are the agitation speed and the suspension volume. To validate the performance of the parameters, the PE experiment was divided into several sub-experiments with varying agitation speeds (600, 800, 1000 rpm) and suspension volume (60 mL, 80 mL, 100 mL). For each sub-experiment, at least three batches were conducted; see Table 3. Each batch consisted of six agitated bottles and one non-agitated control bottle as replicates. The general experimental setup is shown in Figure 1.

The following procedure was conducted:Culture preparation(a)The medium was prepared in a 1 L bottle according to Table 2, before being sealed and flushed with argon for 10 min. The bottle was then stored inside the AC.(b)Seven serum bottles were sterilized by autoclaving (121 °C, 2 bar) for 20 min and then placed in the AC.(c)As a starter, an inoculation culture (IC) was previously cultured in a glass bottle, incubated until fully grown (OD_600_ = 0.15) and placed in the AC before use. It should be noted that the IC temperature should be room temperature.(d)The medium was filled in the serum bottles. Before adding IC, a 50 g/L Na_2_S·9 H_2_O stock solution was added to each bottle (1:100) and shaken to completely reduce the medium.(e)IC was added and a magnetic stirrer rod was placed in each bottle. The bottles were sealed with a rubber stopper and ejected from the AC. The pH was not adjusted. The ratio of IC and medium was 1:2. The suspension volume was altered for each sub-experiment.(f)With a special gas distribution station, the bottles were vacuumed (Vacuubrand VP 100 C) to 300 mbar and pressurized with the 80/20 hydrogen/carbon dioxide mixture. This routine was repeated two more times and the final pressure in the bottles was adjusted to a value of 5 bar absolute.Sub-experiment(a)Six of the seven bottles were placed on the magnetic stirrer inside the incubator, and one bottle was placed non-agitated as a control sample. The agitation speed was adjusted for each sub-experiment.(b)The pressure of one random agitated bottle was recorded.(c)The bottles were incubated for 3 h at 63 °C and then the pressure of each bottle was measured directly after removal from the incubator at incubation temperature.(d)For the next batch, step 1f was repeated.

#### 3.3.2. Reducing Agent Experiment (RE)

The culture preparation (step 1) for RE remained nearly the same, but the reducing agent and its concentration were changed. The reducing agent was freshly prepared for each concentration before addition to the medium. The reducing agent solution was not sterilized. Each variation was carried out in seven consecutive batches to gather a chronological progression. For RE, optimal cultivation conditions were chosen according to the PE results. The overall experimental setup is shown in Figure 1.

### 3.4. Measurement Methods

The serum bottle pressure (absolute) was measured using a pressure sensor combined with a luer-lock adapter and a canula. The measured pressure data were transferred to a desktop PC and stored in a database. The sensor (MSD 6 BAE), device (GMH 5130) and software (EBS 20M 1.6) were obtained from GHM Messtechnik GmbH, Remscheid, Germany. For the RE experiment, the pH (HD 2156.1 with the electrode GE 100 BNC from GHM Messtechnik GmbH, Germany) and ORP (HD 2156.1 with the electrode GR 105 BNC from GHM Messtechnik GmbH, Germany) were measured in the AC at room temperature. ORP measurements during experiments often cannot exclude some possible poisoning of the electrodes. However, the electrode measurement was checked before and after the experiment at room temperature with a calibration solution (GRP 100, bought from GHM Messtechnik GmbH, Germany) to exclude any large deviations.

### 3.5. Calculation of the Methane Evolution Rate

The pressure of one random bottle was recorded during incubation and the pressure before ($p_{1}$) and after ($p_{2}$) incubation in the bottles was measured. The pressure difference was calculated as well as the incubation duration ($d_{t}$). The partial pressure for methane $p_{CH4}$ can be calculated according to Equation (Equation 1) as a result of stochiometric conversion. This method was adapted according to Taubner et al [33]. According to the ideal gas law, the mole number of methane $n_{CH4}$ can be calculated (see Equation (Equation 2)) with $R=0.083144722[\frac{bar*L}{mol*K}]$, T=336.15K. The volume of the gas phase $V_{g}$ is the difference between the known volume of the 118 mL bottle and the suspension volume, being 60, 80 or 100 mL. Finally, the methane evolution rate (MER) was calculated by dividing the amount of methane substance by the batch time and the reactor volume (see Equation (Equation 3)).
(1)$$p_{CH4}=\frac{p_{1}-p_{2}}{4}$$
(2)$$n_{CH4}=\frac{p_{CH4}*V_{g}}{R*T}*1000$$
(3)$$MER=\frac{n_{CH4}}{d_{t}*V_{r}}\;\left[\frac{\mathrm{mmol}}{L*h}\right]$$

## 4. Results

### 4.1. Parameter Experiment

#### 4.1.1. Influence of Agitation Speed and Suspension Volume

The optimal cultivation conditions for *M. marburgensis* in 120 mL bottles were determined by varying the process parameters agitation speed and suspension volume. The results can be seen in Figure 2 and Table 4 with mean values and standard deviation.

The highest mean MER value of 3.382 mmol/L·h was achieved with a setting of 60 mL suspension and 800 rpm. The ratio of suspension to gas phase is exactly 51:49. The ratio for 80 mL is 68:32 and for 100 mL 85:15. Control values vary in the same range, from a mean of 0.333 ± 0.046 to 0.530 ± 0.106 mmol/L·h, indicating that non-agitated conditions result in a much lower MER.

The MER increases with higher agitation speeds as the agitation vortex increases the surface and mass transfer between the suspension and gas phase. However, for 60 mL, there is a significant MER drop from 800 rpm to 1000 rpm, while for 80 and 100 mL it was not observable. The agitation speed of 1000 rpm was also the maximum for bottles filled with 60 mL. Otherwise, the agitation vortex would have disturbed the magnetic stirrer rod and interrupted the agitation. In contrast, for 100 mL, the increase in MER is very low in terms of increasing the agitation speed, but with reproducible values.

All values for 3 h of cultivation were statistically analyzed with Welch’s *t*-test (two-sided, unequal variance), which can be seen in Figure 3. In all cases, there was a significant difference (*p* < 0.05) between the agitation speeds, which shows that the agitation speed has a significant impact on the MER.

#### 4.1.2. Cultivation Duration

In every batch, the pressure of one random agitated bottle was recorded over time. From the record, it is possible to observe how the MER changes with a shorter incubation time, in this case after 2 h and 1 h.

The results in Figure 4 show that the MER decreases for 60 mL as the agitation speed increases from 800 to 1000 rpm. For 80 mL, the same behavior can be observed, as for a 3 h cultivation, the MER increases with increasing agitation speed. It can be seen that with less cultivation time, the differences between the different suspension to gas ratios are not as pronounced. Especially after 1 h, the MER for 100 mL is on a similar level compared to 60 and 80 mL at lower agitation speeds. Higher agitation speeds, as well as a higher suspension to gas ratio, have more influence on MER because a larger agitation vortex can be achieved and therefore the surface area for mass transfer increases. However, the highest mean MER of 4.033 ± 0.538 mmol/L·h was achieved for 80 mL with 1000 rpm after a 1 h cultivation, which was also the highest value obtained throughout the experiment.

### 4.2. Reducing Agent Experiment

#### 4.2.1. Reducing Mechanism

The reducing mechanism of sodium dithionite and sodium sulfide is shown in Equations (Equation 4) and (Equation 5) [34] and Equations (Equation 6) and (Equation 7), respectively. In an alkaline solution, sodium dithionite reacts with oxygen, oxidizing to sulfite and sulfate ions. Furthermore, the formed sulfite ion enables a second reducing step by being further oxidized to sulfate.
(4)$$S_{2}O_{4}^{2-}+O_{2}+2{\mathrm{OH}}^{-}\to {\mathrm{SO}}_{3}^{2-}+{\mathrm{SO}}_{4}^{2-}+H_{2}O$$
(5)$${\mathrm{SO}}_{3}^{2-}+\frac{1}{2}O_{2}\to {\mathrm{SO}}_{4}^{2-}$$

For sodium sulfide, the reducing step is accomplished via a second reaction. First, in contact with water and weak acids, such as carbon dioxide, sodium carbonate and hydrogen sulfide are formed (see Equation (Equation 6)). Second, hydrogen sulfide serves as a source of sulfur for the microorganisms, but can also be oxidized to pure sulfur, hence reducing the medium (see Equation (Equation 7)). However, it may also be possible that hydrogen sulfide is oxidized to sulfuric or sulfurous acid.
(6)$${\mathrm{Na}}_{2}S+{\mathrm{CO}}_{2}+H_{2}O\to {\mathrm{Na}}_{2}{\mathrm{CO}}_{3}+H_{2}S$$
(7)$$2H_{2}S+O_{2}\to 2S+2H_{2}O$$

The reaction time difference of the three different reducing agents was determined with sodium resazurin as a redox potential indicator. Each was added to medium in separate serum bottles (40 mL medium plus 400 μL of a 50 g/L reducing agent solution). The experiment was carried out in an AC and for each color change a picture was taken and the time was measured (see Figure 5). The pH in all bottles was alkaline.

The reducing mechanism of sodium dithionite is very fast. After sodium dithionite was added, the medium immediately became colorless (about 5 s). With sodium sulfide, the medium color turned pink after 20 s, which was slightly lighter after 5.5 min and finally turned colorless after 12 min. The reducing mechanism of L-Cys-HCl took much longer. After 12 min, it could be seen that the color changed to pink, which slowly turned colorless after 35 min, but was still to a certain extent pink. Finally, after 47 min, the medium with L-Cys-HCl was colorless.

#### 4.2.2. Comparison of Different Reducing Agents

Different reducing agents (sodium dithionite, sodium sulfide, L-Cys-HCl) in different concentrations (0.5, 0.25 and 0.1 g/L) were compared at optimal cultivation conditions of 800 rpm, 60 mL suspension volume and 63 °C. For each batch, control bottles were incubated with the same reducing agent and concentration. The exact results can be seen as median values with upper and lower limits in Table 5. All values of the experiment are plotted in Figure 6 as a box plot with the median as a bar.

The highest median MER of 3.884 mmol/L·h was achieved for 0.25 g/L sodium sulfide, followed by 3.855 mmol/L·h (0.1 g/L sodium dithionite), 3.847 mmol/L·h (0.1 g/L L-Cys-HCl) and 3.833 mmol/L·h (0.25 g/L sodium dithionite).

The closest upper and lower limits among all reducing agents and concentrations were achieved for 0.25 g/L sodium dithionite and 0.1 g/L L-Cys-HCl. Overall, L-Cys-HCl with a concentration of 0.25 g/L performed poorly, with a median MER of 1.774 mmol/L·h. The cultivation with the same concentration was repeated with a similar result.

The statistical analysis with Welch’s *t*-test shows that there is a significant difference between 0.5 g/L sodium sulfide and 0.25 g/L sodium dithionite, as well as 0.1 g/L L-Cys-HCl (*p* < 0.0001). However, there is no significant improvement if L-Cys-HCl is used instead of sodium dithionite (*p* = 0.81).

During the experiments, it was observable that black particles precipitated, especially for sodium sulfide but also for sodium dithionite. The visible amount did not increase with higher concentration. The precipitate was not analyzed.

#### 4.2.3. MER Change during Batch Progress

In contrast to the summarized data shown above for each reducing agent, the data for each batch are plotted in Figure 7 as median to interpret the change in MER during consecutive batches. It can be seen that the MER decreases with increasing batch counts, which means that a long incubation time will result in a decrease in the MER. Especially for sodium sulfide but also for 0.5 g/L sodium dithionite and 0.25 g/L L-Cys-HCl, a descent in MER is visible. On the contrary, 0.25 g/L sodium dithionite and 0.1 g/L L-Cys-HCl kept a stable MER throughout the experiment.

#### 4.2.4. pH and Redox Potential

Before and after each sub-experiment, the pH and the redox potential were measured. The results can be seen in Figure 8 and Figure 9 with single data points and the mean values as bars.

The medium was not adjusted to a certain pH before or after the addition of inoculum. Therefore, the pH was only influenced by the addition of the reducing agent and the inoculum. For sodium sulfide, the pH stays at a similar level for all concentrations, whereas differences can be observed for the other reducing agents. A high peak was observed for 0.25 g/L sodium diothionite. A possible explanation may be that the added inoculum was not at room temperature, therefore leading to a false pH value. The same could be applied for 0.25 g/L L-Cys-HCl, as one would expect a pH decrease with an increasing amount of L-Cys-HCl. After the sub-experiment, the pH for all reducing agents commuted at nearly the same stage with a pH of 6.8 to 7.3.

The redox potential of the bottles was measured after the addition of reducing agent and inoculum to the medium, as well as after the sub-experiment. For sodium sulfide, the redox potential was around −416 to −440 mV, which decreased to −380 mV. The bottles with L-Cys-HCl remained nearly at the same level before and after the sub-experiment. In contrast, the reducing potential of sodium dithionite with around −700 mV is very high compared to the other reducing agents. After the sub-experiment, the measured redox potential was in a similar stage as for the other reducing agents, except for 0.5 g/L sodium dithionite, which varied around a mean value of −519 mV.

## 5. Discussion

### 5.1. Cultivation Conditions

This study showed that the proposed method is suitable for determining the optimal cultivation conditions by varying the agitation speed and suspension volume for anaerobic archaea. The standard deviation of the data was quite low and the Welch’s *t*-test showed that the varied cultivation conditions had a significant impact on the MER. The highest MER was achieved with 800 rpm and 60 mL of suspension in 118 mL of serum bottles for *M. marburgensis*. However, if the suspension volume is low (e.g., for 60 mL), faster agitation induces shear forces on the microorganisms resulting in a lower MER.

The duration of cultivation is another important factor, as a high suspension volume (e.g., 100 mL) results in a fast conversion of a low amount of gas in a shorter time due to the high suspension to gas volume ratio (85:15). With a shorter cultivation duration, the MER for 100 mL is similar compared to the MER of 60 and 80 mL. However, the deviation of the values increases with a shorter cultivation time. Therefore, 3 h cultivation is preferred as the measured results are more consistent and less errors can occur.

Furthermore, it is possible to compare different organisms without the use of gas chromatography because the calculated MER provides a suitable comparison factor. Only a pressure sensor and the ideal gas law were used to calculate the MER values. Therefore, the method is very simple and inexpensive to perform and can be used in field experiments or less equipped laboratories.

### 5.2. Usage of Reducing Agents

The use of alternative reducing agents instead of sodium sulfide for *M. marburgensis* is promising, as several advantages can be drawn:A lower reducing agent concentration performed better than a higher concentration. Surprisingly, 0.5 g/L sodium sulfide performed poorly, although it is the state of the art to use this reducing agent and concentration.The MER remained stable in several batches with the use of sodium dithionite, compared to sodium sulfide, which decreases over time. This could also be related to the high redox potential of sodium dithonite and the chemical stability in an alkaline or neutral solution for several weeks, as stated by Telfeyan et al. [35].pH and redox potential showed that the medium used is well buffered to maintain a longer incubation time. However, a medium change is needed due to the formation and overload of surface-active metabolites, such as proteins that lead to foam formation, especially for bubble-column bioreactors [36].For sodium sulfide and sodium dithionite, a precipitate was observed, which sedimented in the bottles. This could be related to the high redox potential of both reducing agents. This high potential may lead to the binding of oxygen from functional groups or salts. However, a correlation between MER change and precipitation could not be drawn, but it may be that the high redox potential has influence on the organisms.Overall, sodium dithionite and L-Cysteine-HCl performed similarly with respect to MER, which was also observed by Mylroie et al. [11]. However, the redox potential of sodium dithionite is higher and does not influence the pH level as much as L-Cys-HCl. Furthermore, the reducing time of L-Cys-HCl is very high compared to sodium dithionite, which leads to a waiting time before the inoculum can be added to the medium.The use of resazurin can be omitted because the redox potential stays at the same stage after several batches. As the medium is frequently changed, it is ensured that the redox potential remains at this level.Sodium dithionite is a much more sustainable and non-toxic chemical (H251, H302) than sodium sulfide (H302, H311, H314, H400). Furthermore, sodium dithionite reduces much faster than sodium sulfide and therefore allows faster process setup and a shorter delay period, which was also observed by Widdel et al. [19].

For further research, other reducing agents can be investigated and applied for biological methanation. It is important to note that an external source of sulfur, such as sulfite or hydrogen sulfide, is needed. This is provided by adding a reducing agent, as the use of sulfate ions (${\mathrm{SO}}_{4}^{2-}$) is not suitable for the cultivation of *M. marburgensis* [29,37]. Furthermore, the (long-term) use of sodium dithionite and lower concentrations to prevent precipitation will be further investigated and reported.

### 5.3. Economic Considerations

Besides the MER performance of the reducing agents, economics should also be considered. With respect to the scale-up of biological methanation reactors, larger amounts of chemicals need to be supplied. For example, with a concentration of 0.25 g/L and a reactor of 10 m³, an amount of 2.5 kg of reducing agent is needed. In Table 6, the costs of the three used reducing agents for different suppliers are shown. The prices are changing due to quality, producer, amount and energy prices. However, it can be seen that L-Cys-HCl is a highly priced chemical, whereas sodium dithionite has the lowest cost, around EUR 60 to 70 per kilogram, and sodium sulfide is twice the price. In the future, a life cycle assessment of the chemicals used for biological methanation should be performed to provide further information on the production and treatment of waste water for such chemicals.

## Figures and Tables

**Figure 1 microorganisms-11-02533-f001:**
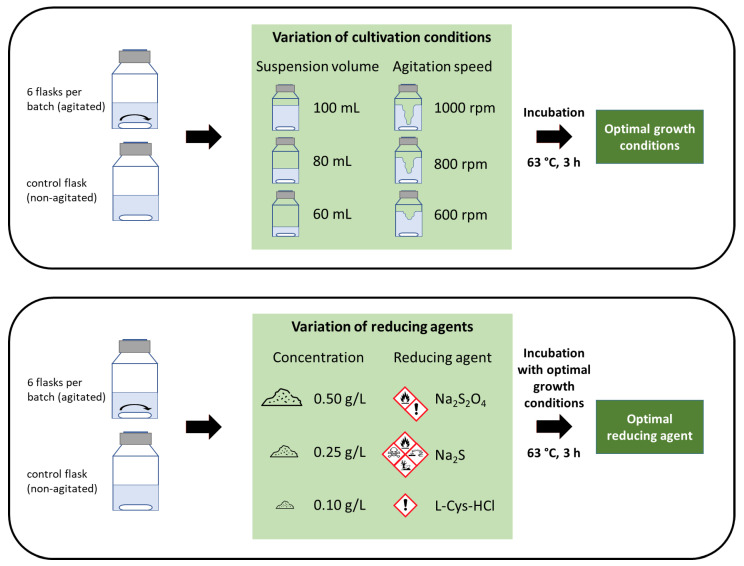
The figure shows the overall experimental setup with the PE (**upper part**) and the RE (**lower part**).

**Figure 2 microorganisms-11-02533-f002:**
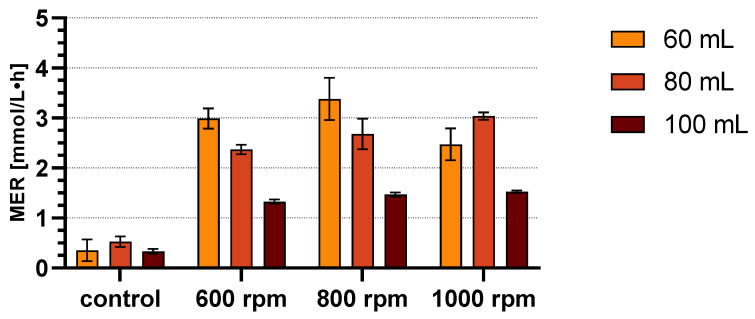
Results of the parameter variation experiment for 3 h cultivation. The MER values are plotted as bars with mean and standard deviation. Exact values can be taken from Table 4.

**Figure 3 microorganisms-11-02533-f003:**
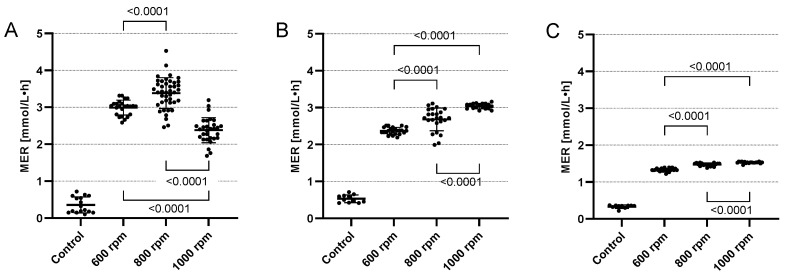
Comparison of the different agitation speeds with Welch’s *t*-test for 60 mL (**A**), 80 mL (**B**) and 100 mL (**C**). The MER values are plotted as scattered points with mean as bar and standard deviation.

**Figure 4 microorganisms-11-02533-f004:**
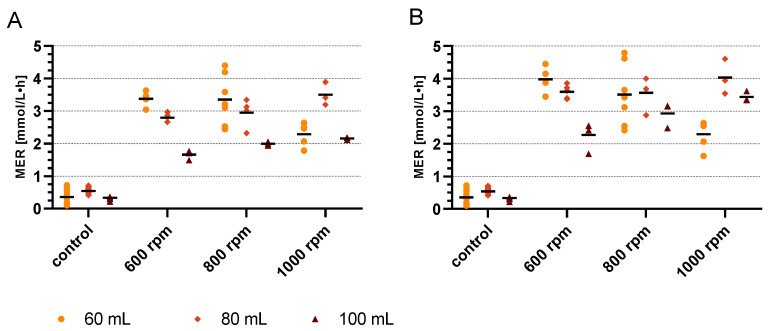
Results of the parameter variation experiment for 2 h (**A**) and 1 h (**B**) cultivation. The MER values of each suspension volume were plotted as aligned circles (60 mL), rhombs (80 mL) and triangles (100 mL). The mean MER value is shown as a bar.

**Figure 5 microorganisms-11-02533-f005:**
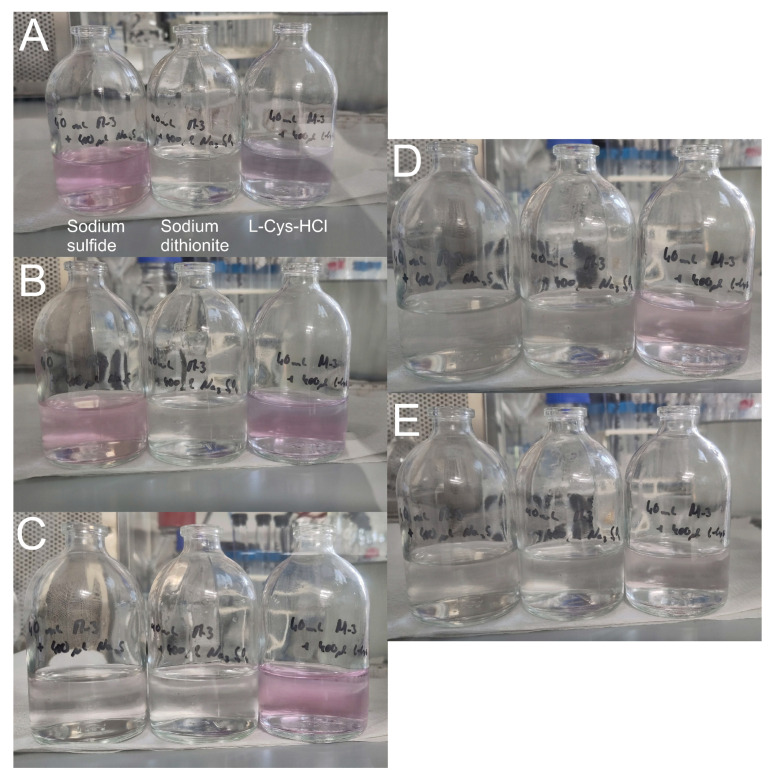
Color change of sodium resazurin in medium with different reducing agents. The pictures were taken consecutively and the time was measured: (**A**) 20 s, (**B**) 5.5 min, (**C**) 12 min, (**D**) 35 min, and (**E**) 47 min. The pH of the medium with sodium sulfide was 9.04, with sodium dithionite 8.71 and with L-Cys-HCl 8.23.

**Figure 6 microorganisms-11-02533-f006:**
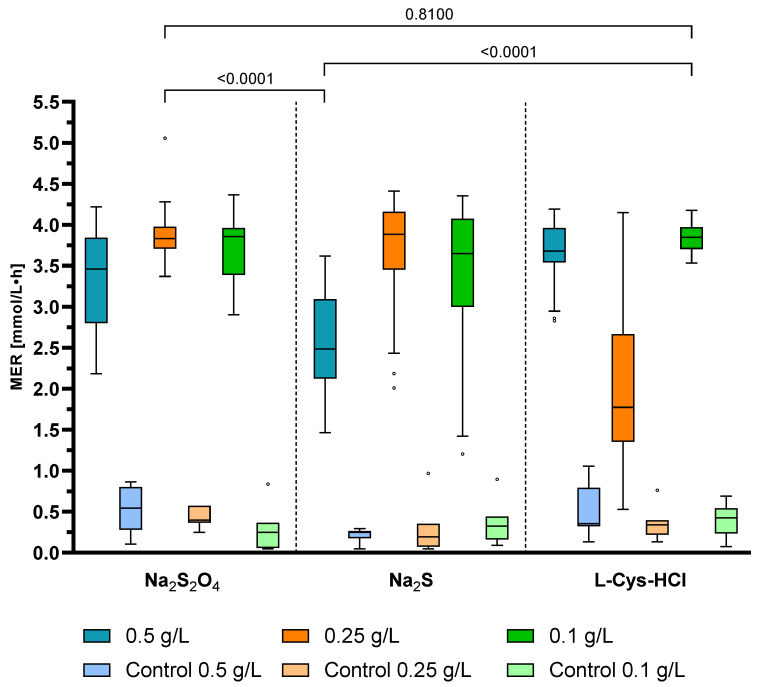
Results of the reducing agent variation experiment with plotted MER of the different concentrations. For each reducing agent and concentration, a non-agitated sample with the same reducing agent and concentration was measured as a control value (see Figure 1). The incubation was performed with the determined optimal cultivation conditions of 800 rpm agitation, 60 mL suspension volume and 63 °C. The boxes extend from the 25th to 75th percentiles with the bar as the median. The whiskers are plotted according to the Tukey method. Data beyond the whiskers are marked as outliers.

**Figure 7 microorganisms-11-02533-f007:**
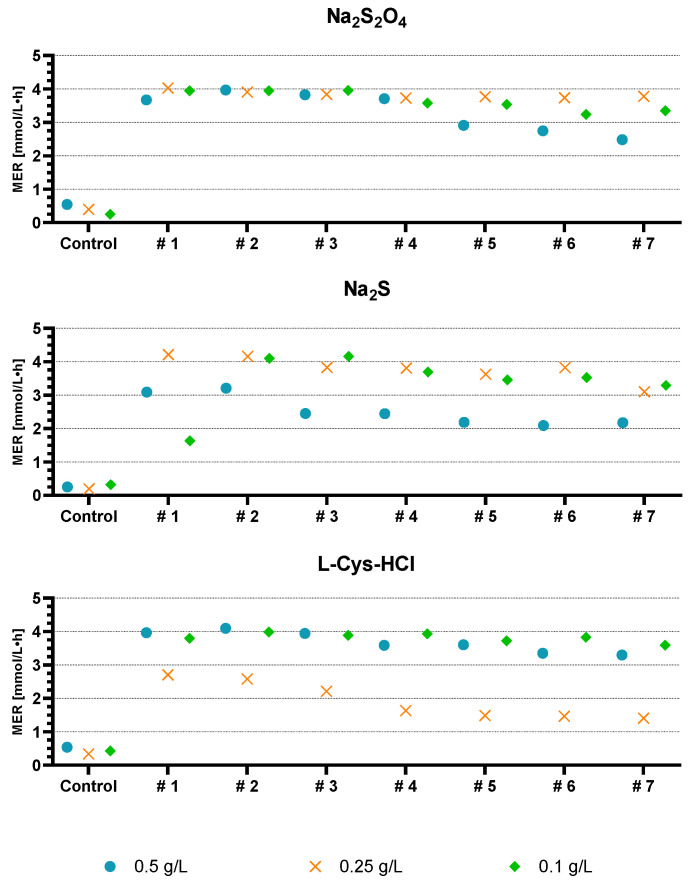
MER progression over consecutive batches. The graphs show each reducing agent with their respective concentration of 0.5 g/L (blue circle), 0.25 g/L (orange cross) and 0.1 g/L (green rhombus). The values shown are the medians of each batch.

**Figure 8 microorganisms-11-02533-f008:**
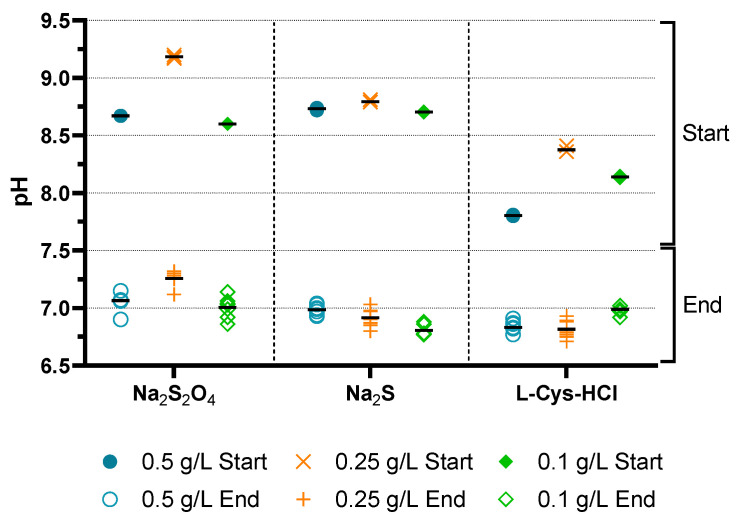
Plot of pH values before and after the experiment. The graphs show each reducing agent with their respective concentration of 0.5 g/L (blue circle), 0.25 g/L (orange cross) and 0.1 g/L (green rhombus). For a better overview, the start and end values are superimposed for each concentration. The mean value is shown as a bar.

**Figure 9 microorganisms-11-02533-f009:**
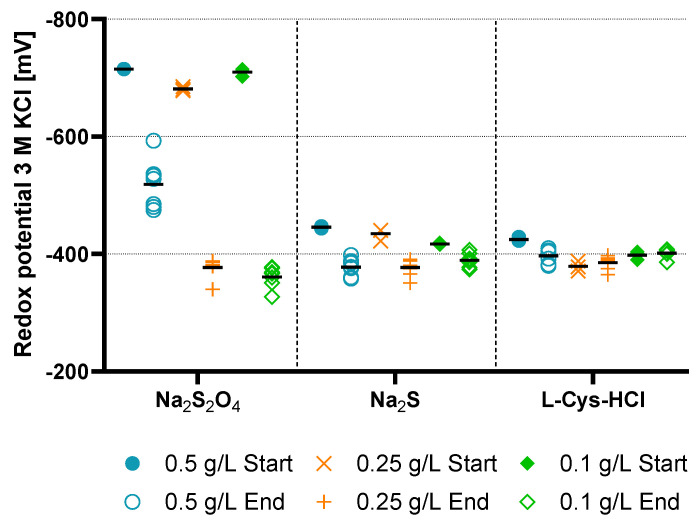
Plot of the redox potential values of a 3 M KCl electrode in mV before and after the sub-experiment. The graphs show each reducing agent with their respective concentration of 0.5 g/L (blue circle), 0.25 g/L (orange cross) and 0.1 g/L (green rhombus). The mean value is shown as a bar.

**Table 1 microorganisms-11-02533-t001:** Overview of various reducing agents with their respective ORP value $E^{0}$, which is the standard redox potential based on a standard hydrogen electrode for a solution with equal concentrations of reductants and oxidants [15]. The superscript letters after individual values refer to the corresponding reference.

Chemical	Concentration	*E*^0^ [mV]	Reference
Na_2_S·9H_2_O	0.25–0.50 g/L	−243 ${}^{a}$ /−571 ${}^{b}$	[15,16] ${}^{a}$, [17] ${}^{b}$
Cysteine/Cystine	0.25–0.50 g/L	−325 ${}^{c}$/−340 ${}^{d}$	[15,16,18] ${}^{c}$, [17] ${}^{d}$
2 Sulfit^2−^/Dithionite^2−^	10–30 mg/L	−574	[18,19]
Dehydroascorbate/Ascorbate	0.50–1.0 g/L	+58	[15,16,17,18]
Dithiothreitol	1 mM, 0.1–0.5 g/L	−330	[17,18,20]
FeS (amorphous hydrated)	11 µg/mL	−270	[15,21]
Sulfite (SO_4_${}^{2-}$/SO_3_${}^{2-}$)		−516	[16,22]
Dithioglycolate/Thioglycolate	0.50–1.0 g/L	−140	[15,16,17,18]
Titanium(III)citrate	1–4 mM	−480	[15,23,24]
H_2_ (PdCl_2_)	Variable	−413	[15]

**Table 2 microorganisms-11-02533-t002:** Medium composition used for the cultivation of *M. marburgensis*.

Component	Amount	Unit
NH_4_Cl	2.1	g/L
K_2_HPO_4_	6.8	g/L
TES	1	mL/L
Na-resazurin solution (0.1% *w*/*v*)	0.5	mL/L
1 M Na_2_CO_3_ solution	6	mL/L

**Table 3 microorganisms-11-02533-t003:** Setup and number of conducted batches for the parameter experiment. The overall number of agitated bottles plus non-agitated control bottles is in brackets.

		Volume (mL)		
		**60**	**80**	**100**
Agitation speed (rpm)	600	4 (24 + 4)	4 (24 + 4)	4 (24 + 4)
	800	7 (42 + 7)	4 (24 + 4)	3 (18 + 3)
	1000	5 (30 + 5)	3 (18 + 3)	3 (18 + 3)

**Table 4 microorganisms-11-02533-t004:** Data of the MER cultivation for 3 h in mmol/L·h as mean values with standard deviation.

	Volume [mL]		
	**60**	**80**	**100**
control	0.357 ± 0.217 (n = 16)	0.530 ± 0.106 (n = 11)	0.333 ± 0.046 (n = 10)
600 rpm	2.991 ± 0.203 (n = 24)	2.370 ± 0.094 (n = 24)	1.328 ± 0.041 (n = 24)
800 rpm	3.382 ± 0.419 (n = 42)	2.681 ± 0.307 (n = 24)	1.473 ± 0.042 (n = 18)
1000 rpm	2.473 ± 0.318 (n = 18)	3.040 ± 0.073 (n = 18)	1.528 ± 0.003 (n = 18)

**Table 5 microorganisms-11-02533-t005:** MER values for the different reducing agents and respective concentrations of 0.5, 0.25 and 0.1 g/L as median values, with the upper and lower limits in brackets. The control values are displayed underneath in the same manner.

Median MER values with upper and lower limits in brackets (mmol·L^−1^ ·h^−1^)			
	**0.5 g/L**	**0.25 g/L**	**0.1 g/L**
${\mathrm{Na}}_{2}S_{2}O_{4}$	3.459 (3.844, 2.799)	3.833 (3.976, 3.708)	3.855 (3.961, 3.389)
${\mathrm{Na}}_{2}S$	2.484 (3.092, 2.125)	3.884 (4.162, 3.452)	3.649 (4.074, 2.997)
L-Cys-HCl	3.679 (3.961, 3.540)	1.774 (2.667, 1.352)	3.847 (3.972, 3.704)
**Control median MER values with upper and lower limits in brackets (mmol·L** ${}^{-1}\cdot$ **h** ${}^{-1}$ **)**			
	**0.5 g/L**	**0.25 g/L**	**0.1 g/L**
${\mathrm{Na}}_{2}S_{2}O_{4}$	0.542 (0.799, 0.279)	0.396 (0.572, 0.366)	0.249 (0.366, 0.059)
${\mathrm{Na}}_{2}S$	0.249 (0.264, 0.176)	0.191 (0.352, 0.073)	0.322 (0.440, 0.161)
L-Cys-HCl	0.352 (0.791, 0.322)	0.337 (0.396, 0.220)	0.425 (0.542, 0.234)

**Table 6 microorganisms-11-02533-t006:** Costs for the reducing agents per kilogram. The prices were retrieved online in the respective webshop on 15 September 2023.

	Sodium Dithionite (≥95%)	Sodium Sulfide (≥98%)	L-Cysteine-HCl (≥98%)
Sigma-Aldrich	EUR 60.6 (1065051000)	EUR 126.4 (S2006-500G)	EUR 597 (C7880-1KG)
VWR	EUR 60.6 (1.06505.1000)	EUR 166 (36622.A1)	EUR 466 (1.02839.1000)
FisherScientific	EUR 69.56 (10274490)	EUR 132.2 (10587952)	EUR 488.8 (11478643)

## Data Availability

The data presented in this study are available on request from the corresponding author. The data are not publicly available due to privacy.

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
