# Peer review of "Comparison of Various Reducing Agents for Methane Production by Methanothermobacter marburgensis"

_microorganisms, 2023, doi:10.3390/microorganisms11102533_

Round 1
Reviewer 1 Report
The manuscript presents a robust methodology for evaluating the success of specific experimental conditions for the growth of methane-producing microorganisms. The authors' contribution is particularly noteworthy due to its practical implications for experimental methods. Dealing with strict anaerobic microorganisms typically demands time-intensive and costly techniques. The development of alternative experimental methods not only saves time and improves laboratory results but also holds significant economic implications. The authors briefly touch upon these implications, especially concerning the scalability of the technology, which adds depth to their research findings.
Author Response
Dear reviewer, thank you very much for taking your time to read the manuscript and for your nice feedback.
Reviewer 2 Report
microorganisms-2640237: Review & Recommendations
Comments and Suggestions for Authors
1. Title.
Please change the title for more precise one: “Comparison of various reducing agents for methane production by Methanothermobacter marburgensis”.
2. Abstract.
Lines 4-7 (“Until now, sodium sulfide (Na2S) has been used mainly for this purpose based on earlier published articles at the beginning of anaerobic microbiology research. In this study, the usage of alternative reducing agents like sodium dithionite (Na2S2O4) and L-Cysteine-HCl shows that similar results can be obtained with less environmental and hazardous impacts.”) produce some impression that the authors are the first ones which reduce media with dithionite and cystein. To avoid this impression, please, add a few words: “In continuation of the earlier investigations, in this study, the usage of alternative reducing agents…”.
3. Keywords.
Technical misprints – please change for capital letters: l-cysteine-HCl; Methanothermobacter marburgensis.
4. Introduction + State of art.
Selection of reducing agents for methanation has a long and interesting history. The sections “Introduction” and “State of art” present the most popular examples. However, I ask to add, at least, two following classical methods.
1) Electrochemical method (reduction with cathode hydrogen): Hanke, M.E. and Katz, Y.J. An electrolytic method for controlling oxidation-reduction potential and its application in the study of anaerobiosis. Arch. Biochem. Biophys. 1943. 2: 183-200. This method permits to grow strictly anaerobic mathanogens without additional reagents.
2) Reduction with insoluble ferrous sulfide sludge: Brock, T.D. and Od'ea K. Amorphous ferrous sulfide as a reducing agent for culture of anaerobes. Applied and Environmental Microbiology.1977. 33 (2): 254-256. DOI: https://doi.org/10.1128/aem.33.2.254-256.1977
By the way, the last method is a working one in the DSMZ till present (https://bacmedia.dsmz.de/solutions/2526). As far as the ferrous sulfide sludge is insoluble, it is not corrosive for steel and could be used for industry.
5. Materials and Methods.
5.1.
Authors used the platinum electrode GR 105 BNC (Line 180). The platinum electrodes can be poisoned with reduced sulfur compounds. See, for example:
https://www.sciencedirect.com/science/article/pii/S1572665718301826 or https://arxiv.org/ftp/arxiv/papers/1108/1108.0609.pdf
Thus, there is some suspicion that the redox data could be not very punctual. The basis for this suspicion: the maximal redox value for sodium dithionite is -660 mV while the authors presented -700 mV as a measured value (Line 300).
Normally, researchers check their electrodes during experiments with intermediate checking calibration with the ZoBell solution (See: https://in-situ.com/en/zobells-orpredox-calibration-solution
or https://www.coleparmer.com/i/ysi-3682-zobell-solution-for-orp-calibration-1-25-g/0547860 etc).
All these doubts could be excluded if authors did not limit their redox color indication with sodium resazurin (Line 113). There are some other – not so popular - color redox indicators which also are not toxic for microorganisms. For example, the redox indicator 1,1'-dibenzyl-4,4'-bipyridinium dichloride (= benzyl viologen) change its color at -359 mV (at pH 7.0).
The work has already been finished. So, there is no sense to repeat the experiments for so small improvement.
Thus, I ask authors:
- for future research - to mind my information for experiments;
- for present article - to add a few phrases, for example: “Often EPR measurements during experiments cannot exclude some possible poisoning of the electrodes. As far as there were no additional calibrations of the electrodes during the experiments, the measured redox values could be slightly overestimated in the negative direction.”
5.2.
The paper contains essential information about the medium composition and reducing agents. Bottles with the medium were sterilized at high temperature (Line 142). However, there is no information about methods of sterilization for reducing agents. Please add some short description of their sterilization or mention that they were not sterile (Line 169).
6. Results.
No questions.
7. Discussion.
No questions.

Author Response
Dear reviewer,
First of all, thank you very much for taking your time to read our manuscript. The scientific exchange and discussion add a lot of value to this manuscript and our knowledge. We’ve addressed all of your comments and suggestions as can be seen in the PDF file, marked with red color. The changes made in the manuscript are also marked and attached to the PDF file.
